

# Mitochondrial genomes organization in alloplasmic lines of sunflower (*Helianthus annuus* L.) with various types of cytoplasmic male sterility

Maksim S. Makarenko[1,*], Igor V. Kornienko[1,2], Kirill V. Azarin[1,*], Alexander V. Usatov[1], Maria D. Logacheva[3], Nicolay V. Markin[1] and Vera A. Gavrilova[4]

[1] Southern Federal University, Rostov-on-Don, Russia
[2] Southern Scientific Center of the Russian Academy of Sciences, Rostov-on-Don, Russia
[3] Moscow State University, Belozersky Institute of Physical and Chemical Biology, Moscow, Russia
[4] The N.I. Vavilov All Russian Institute of Plant Genetic Resources, Saint Petersburg, Russia
* These authors contributed equally to this work.

Corresponding author
Maksim S. Makarenko,
mcmakarenko@yandex.ru

## ABSTRACT

**Background:** Cytoplasmic male sterility (CMS) is a common phenotype in higher plants, that is often associated with rearrangements in mitochondrial DNA (mtDNA), and is widely used to produce hybrid seeds in a variety of valuable crop species. Investigation of the CMS phenomenon promotes understanding of fundamental issues of nuclear-cytoplasmic interactions in the ontogeny of higher plants. In the present study, we analyzed the structural changes in mitochondrial genomes of three alloplasmic lines of sunflower (*Helianthus annuus* L.). The investigation was focused on CMS line PET2, as there are very few reports about its mtDNA organization.

**Methods:** The NGS sequencing, *de novo* assembly, and annotation of sunflower mitochondrial genomes were performed. The comparative analysis of mtDNA of HA89 fertile line and two HA89 CMS lines (PET1, PET2) occurred.

**Results:** The mtDNA of the HA89 fertile line was almost identical to the HA412 line (NC_023337). The comparative analysis of HA89 fertile and CMS (PET1) analog mitochondrial genomes revealed 11,852 bp inversion, 4,732 bp insertion, 451 bp deletion and 18 variant sites. In the mtDNA of HA89 (PET2) CMS line we determined 27.5 kb and 106.5 kb translocations, 711 bp and 3,780 bp deletions, as well as, 5,050 bp and 15,885 bp insertions. There are also 83 polymorphic sites in the PET2 mitochondrial genome, as compared with the fertile line.

**Discussion:** The observed mitochondrial reorganizations in PET1 resulted in only one new open reading frame formation (*orfH522*), and PET2 mtDNA rearrangements led to the elimination of *orf777*, duplication of *atp6* gene and appearance of four new ORFs with transcription activity specific for the HA89 (PET2) CMS line—*orf645*, *orf2565*, *orf228* and *orf285*. *Orf228* and *orf285* are the *atp9* chimeric ORFs, containing transmembrane domains and possibly may impact on mitochondrial membrane potential. So *orf228* and *orf285* may be the cause for the appearance of the PET2 CMS phenotype, while the contribution of other mtDNA reorganizations in CMS formation is negligible.

## INTRODUCTION

In plants, the phenomenon of cytoplasmic male sterility (CMS) stems from interaction between mitochondrial and nuclear genomes resulting in microsporogenesis disorders (*Touzet & Meyer, 2014*). All known natural CMSs, as well as most of the artificially obtained examples, are characterized by a special type of mitochondrial DNA (mtDNA) with numerous structural rearrangements as compared to the mtDNA of the fertile plants of the same species (*Ivanov & Dymshits, 2007*; *Horn, Gupta & Colombo, 2014*; *Garayalde et al., 2015*). The mitochondrial genomes of higher plants have comparatively large sizes with a multitude of noncoding and repetitive sequences that can result in the complex of sub-genomic structures (*Chen & Liu, 2014*). These features of plant mtDNA promote a large number of recombination events, leading to the appearance of new sequences and new open reading frames (ORFS), that in turn often results in CMS development (*Horn, Gupta & Colombo, 2014*; *Touzet & Meyer, 2014*).

Natural CMS forms have been described in more than 150 species of flowering plants (*Fujii & Toriyama, 2009*). Most CMS sources in crops are obtained using interspecific hybridization. The first CMS in a sunflower was discovered by *Leclercq (1969)* in an interspecific hybrid between *Helianthus petiolaris* Nutt (PET1) and *Helianthus annuus* L. Comparison of mitochondrial DNA organization of the fertile line and the male-sterile line carrying the PET1 cytoplasm revealed the presence of an 11-kb-inversion and five-kb-insertion (*Siculella & Palmer, 1988*; *Kohler et al., 1991*). These rearrangements of the mitochondrial genome produced a new ORF (*orfH522*) in the 3′-flanking region of the *atp1* gene encoding the alpha subunit of mitochondrial F1 ATPase. A new *orfH522* is co-transcribed with the *atp1* gene as a polycistronic mRNA (*Monéger, Smart & Leaver, 1994*). Using antibodies specific to the product of *orfH522* gene (16-kDa-protein), *Horn et al. (1996)* showed that this was the only difference between the mitochondrial translation products of fertile and CMS lines. The 16-kDa protein is synthesized in all tissues of a plant. It is embedded in the mitochondrial membranes and is believed to disrupt its integrity (*Horn et al., 1996*). Expression of *orfH522* in tapetum cells leads to premature apoptosis. Release of cytochrome C from the mitochondria, activates the proteolytic enzyme cascade, eventually leading to degradation of nuclear DNA and cell death (*Balk & Leaver, 2001*; *Sabar et al., 2003*). Interestingly, stable transgenic CMS tobacco lines carrying the *orfH522* gene were obtained (*Nizampatnam et al., 2009*). When dominant nuclear restorer gene (Rf) is present in the sunflower genome, fertility is restored due to the anther-specific lowering of the co-transcript of *orfH522* and the *atp1* gene (*Monéger, Smart & Leaver, 1994*; *Horn et al., 2003*). A possible mechanism leading to a reduction in the number of the chimeric *atp1-orfH522* transcripts by the Rf is polyadenylation of RNA matrices, causing accelerated degradation of RNA molecules by the ribonuclease (*Gagliardi & Leaver, 1999*).

The CMS-Rf system is widely used for the commercial production of F1 hybrid seeds for many important crops, including maize, sorghum, sunflower etc. (*Liu et al., 2011*;

*Bohra et al., 2016*). Almost all commercial sunflower hybrids are currently based on a single source of CMS discovered by *Leclercq (1969)* and described above. Such genetic homogeneity of cultivated hybrids makes them extremely vulnerable to new virulent strains of the pathogens and can lead to negative phenomena, for example, epiphytotics development (*Levings, 1990*). For instance, leaf blight pandemic affected only one type maize hybrids (namely, Texas-type), while other types of CMS were less susceptible to this disease (*Bruns, 2017*). To create new CMS-Rf systems, prevent mtDNA unification, and reduce genetic vulnerability of sunflower hybrids to biotic and abiotic stresses, it is urgent to search for and introduce the new CMS sources into sunflower breeding. Although more than 70 cytoplasmic male sterility types have been identified in sunflower (*Garayalde et al., 2015*), they have not been sufficiently studied, resulting in limitation of their use in commercial hybrid breeding. Undoubtedly, research on the cytoplasmic male sterility phenomenon is important for investigating the fundamental problem of nuclear-cytoplasmic interaction in the ontogeny of higher plants (*Hanson & Bentolila, 2004*). Previously, the comparison of mitochondrial genome organization between 28 CMS sources of sunflower, performed with Southern hybridization, demonstrated that some types of CMS (for example, ANN2, PET2, PEF1, etc.) have a different organization of the mtDNA from the PET1-like cytoplasms (*Horn, 2002*). Sequencing and comparing of whole mitochondrial genomes of various CMS sources will provide additional information about the molecular changes in their mtDNA, which in turn could help to suggest new mechanisms of the male sterility formation.

In the current study, we investigated structural changes in mitochondrial genomes of HA89-alloplasmic lines: fertile line and two analog lines with different types of cytoplasmic male sterility—PET1 and PET2. The results obtained for the PET2 CMS type formed the basis for further research.

## MATERIALS AND METHODS

### Plant material

Fertile line HA89 and isonuclear CMS lines—PET1 and PET2 of sunflower were obtained from the genetic collection of the N. I. Vavilov Institute of Plant Genetic Resources (VIR, Saint-Petersburg, Russia). The lines had the same nuclear genome (HA89), but they differed in chloroplast and mitochondrial genomes, inherited from their wild ancestors. The CMS sources were initially obtained by the interspecific hybridization of domesticated sunflower (*H. annuus* L.) with *H. petiolaris* Nutt (*Leclercq, 1969*; *Whelan, 1980*).

### Mitochondrial DNA extraction, genome library construction and NGS sequencing

We extracted the organelle fraction with a reduced amount of nuclear DNA from leaves of 14-day sunflower seedlings, following the protocol of *Makarenko et al. (2016)*. For every line, we used the same quantity of leaf tissue from five plants. The DNA isolation was performed with PhytoSorb kit (Syntol, Moscow, Russia), according to the manufacturer's protocol. The NGS libraries preparations were made using one ng of DNA and Nextera XT DNA Library Prep Kit (Illumina, Mountain View, CA, USA), following the sample

preparation protocol by Illumina. For the qualitative control of libraries, Bioanalyzer 2100 (Agilent, Santa Clara, CA, USA) was used. The libraries quantitation was performed with the Qubit fluorimeter (Invitrogen, Carlsbad, CA, USA) and qPCR. Libraries for NGS sequencing were diluted up to the concentration of eight pM. Libraries were sequenced on different sequencing platforms. Fertile line and PET1 NGS libraries were sequenced with NextSeq 500 sequencer using High Output v2 kit (Illumina, Carlsbad, CA, USA). A total number of 13,240,057 150-bp paired reads were generated for fertile line and 14,758,067 reads—for PET1 line. PET2 library was sequenced with HiSeq2000 and MiSeq platforms using TruSeq SBS Kit v3-HS and MiSeq Reagent Kit v2 500-cycles (Illumina, San Diego, CA, USA). A total number of 4,471,774 125-bp and 4,931,318 250-bp paired reads were generated for the PET2 line.

## Analysis of sequencing data

Quality of reads was determined with Fast QC. Trimming of adapter-derived and low-quality (Q-score below 25) reads was performed with Trimmomatic software (*Bolger, Lohse & Usadel, 2014*). Using the Bowtie2 tool v 2.3.3 (*Langmead & Salzberg, 2012*), sequencing reads were aligned to the reference sequence from NCBI databank (NC_023337.1). The Bowtie 2 alignments were done only for concordant paired reads (–no-mixed, –no-discordant options). Variant calling was made with *samtools/bcftools* software (*Li, 2011*) and manually revised using the IGV tool (*Thorvaldsdóttir, Robinson & Mesirov, 2013*). De novo assembly was performed with SPAdes Genome Assembler v 3.10.1 (*Nurk et al., 2013*) with different $K$ values equal to 75, 85, 95, 127, and read coverage cutoff value equal to 30.0 (–cov-cutoff option). The potential ORFs were identified using ORFfinder. The graphical genome map was generated using the OGDRAW tool (*Lohse, Drechsel & Bock, 2007*). Transmembrane domains were predicted using the TMHMM Server v.2.0 (available online: http://www.cbs.dtu.dk/services/TMHMM-2.0/).

## Validation of genome assembly—PCR and Sanger sequencing

The contigs obtained in de novo assembly were aligned to the reference sunflower mitochondrial genome (NC_023337.1) using BLAST. Validation of discovered rearrangements was made by PCR analysis and Sanger sequencing. PCR reactions were performed with LongAmp Taq PCR Kit (New England Biolabs, Ipswich, MA, USA) for reactions with expected amplicons more than 1.5 kb, and with Tersus Plus PCR kit (Evrogen, Moscow, Russia) for other reactions, including Sanger sequencing. For 28–29 cycles of PCR, we used 0.4 uM of primers (Table 1) and one ng of extracted DNA. The direct sequencing of purified amplicons was performed using the BigDye Terminator v3.1 Cycle Sequencing Kit (Thermo Fisher Scientific, Waltham, MA, USA) and ABI Prism 3130xl Genetic Analyser (Applied Biosystems, Foster City, CA, USA).

## RNA extraction and qRT-PCR

Total RNA from the leaves of five samples of each line was extracted with guanidinium thiocyanate-phenol-chloroform reagent kit—ExtractRNA (Evrogen, Moscow, Russia). RNA quality and concentration were measured using the NanoDrop 2000 spectrophotometer

**Table 1 The primers sets used for HA89 (PET2) genome reorganizations validation and gene expression analysis.**

| The purpose | Primer name | Primer sequence (5′-3′) | Line | The expected amplicon size (kbp) | The real amplicon size (kbp) |
|---|---|---|---|---|---|
| Validation sub-genome structure 153.5 kb circle | 189837F | CGTGAAGCCGGGATGGTATT | Fertile | – | 0.9 |
| | 37192R | CAAGTGATCCCCCATCCAGG | PET1 | – | 0.9 |
| | | | PET2 | 0.9 | 0.9 |
| | 189660F | AGGAGTGAGATGGACGCTCT | Fertile | – | 1.8 |
| | 37954R | AAGTGTTGCACCCCCTTGAA | PET1 | – | 1.8 |
| The analysis of *orf645* expression | orf645F | GCCTTCCACCTCTCGTTTGA | Fertile | – | – |
| | orf645R | TCCGAAAGCCGGCCTAAAAT | PET2 | 0.162 | 0.162 |
| The analysis of *orf2565* expression | orf2565F | TCAATCCATGTGTTCTCGCT | Fertile | – | – |
| | orf2565R | CGGAAAGAACAGGTCTCGGT | PET2 | 0.147 | 0.147 |
| The analysis of *atp6* 906/1,056 bp transcripts | atp6F | AGAACTGTAACTGACAACGC | Fertile | 0.106 | 0.106 |
| | atp6R | ACCTGAGTCCGAGTCTGCATC | PET1 | 0.106 | 0.106 |
| | | | PET2 | 0.106 | 0.106 |
| | atp6-1056F | TCCCATGCCTTTCTTGGTCG | Fertile | – | 0.28 |
| | atp6R | -ǁ- | PET1 | – | 0.28 |
| | | | PET2 | 0.28 | 0.28 |
| The analysis of *atp9* 261/300 bp transcripts | atp9F | CATTGGGGCAAACGATGCAA | Fertile | 0.107 | 0.107 |
| | atp9R | CCTCGATTCATTCCGTGGCT | PET1 | 0.107 | 0.107 |
| | | | PET2 | 0.107 | 0.107 |
| | atp9F | -ǁ- | Fertile | – | 0.233 |
| | atp9.300R | TGAAAAAGAAAAAGCGTGAGGAGA | PET1 | – | 0.233 |
| | | | PET2 | – | 0.233 |
| The analysis of *atp1* expression | atp1F | CCCATGGCACAGCCAGAATA | Fertile | 0.14 | 0.14 |
| | atp1R | CAGAAACGCTCAACTGTGGC | PET2 | 0.14 | 0.14 |
| The analysis of *orf285* expression | orf285F | TCCCATCATGACCTACCCGT | Fertile | – | – |
| | atp9.300R | -ǁ- | PET2 | 0.243 | 0.243 |
| Sanger resequencing the 5′ and 3′ ends of 5,050 bp insertion | 274655F | GGTTGAACTAGACCCGCACA | Fertile | – | – |
| | Pet2-seqR | GAAGGAACGAGACAGCACCA | PET2 | 0.7 | 0.7 |
| | Pet2-seqF | AGGGAGAGGGACGAAGTGAC | Fertile | – | – |
| | 275503R | TAACCGCTGCAAGAGTGAGG | PET2 | 0.7 | 0.7 |
| Detection the 5′ and 3′ end of 15.885 bp insertion | 35202F | AGCTCTCCCCATCGGTAGTT | Fertile | – | – |
| | 271194R | GGTCATCAGTTCGAGTGGCA | PET2 | 2.5 | 2.5 |
| | Pet2-insF | AGGAAAAGACCCAACAGGCA | Fertile | – | – |
| | 194675R | TAGCTCTTCCGGAGCACTCT | PET2 | 2.7 | 2.7 |

**Note:**
All PCR reactions were held for HA89 fertile and CMS (PET2) lines. For simplicity primers were named according to their position in the HA89 fertile genome. The "F" and "R" letters denote the PCR strand orientation—forward (plus) and reverse (minus), respectively.

(Thermo Fisher Scientific, Waltham, MA, USA) and the Qubit fluorimeter (Invitrogen, Carlsbad, CA, USA). According to the manufacturer's instruction, 0.5 μg of total RNA was treated with DNAse I (Thermo Fisher Scientific, Waltham, MA, USA). First-strand cDNA was synthesized using MMLV RT kit (Evrogen, Moscow, Russia) and specific primers. The qPCR was performed with designed primers (Table 1) and PCR kit with EvaGreen dye (Syntol, Russia) on Rotor-Gene 6000 (Corbett Research, Mortlake, NSW, Australia).

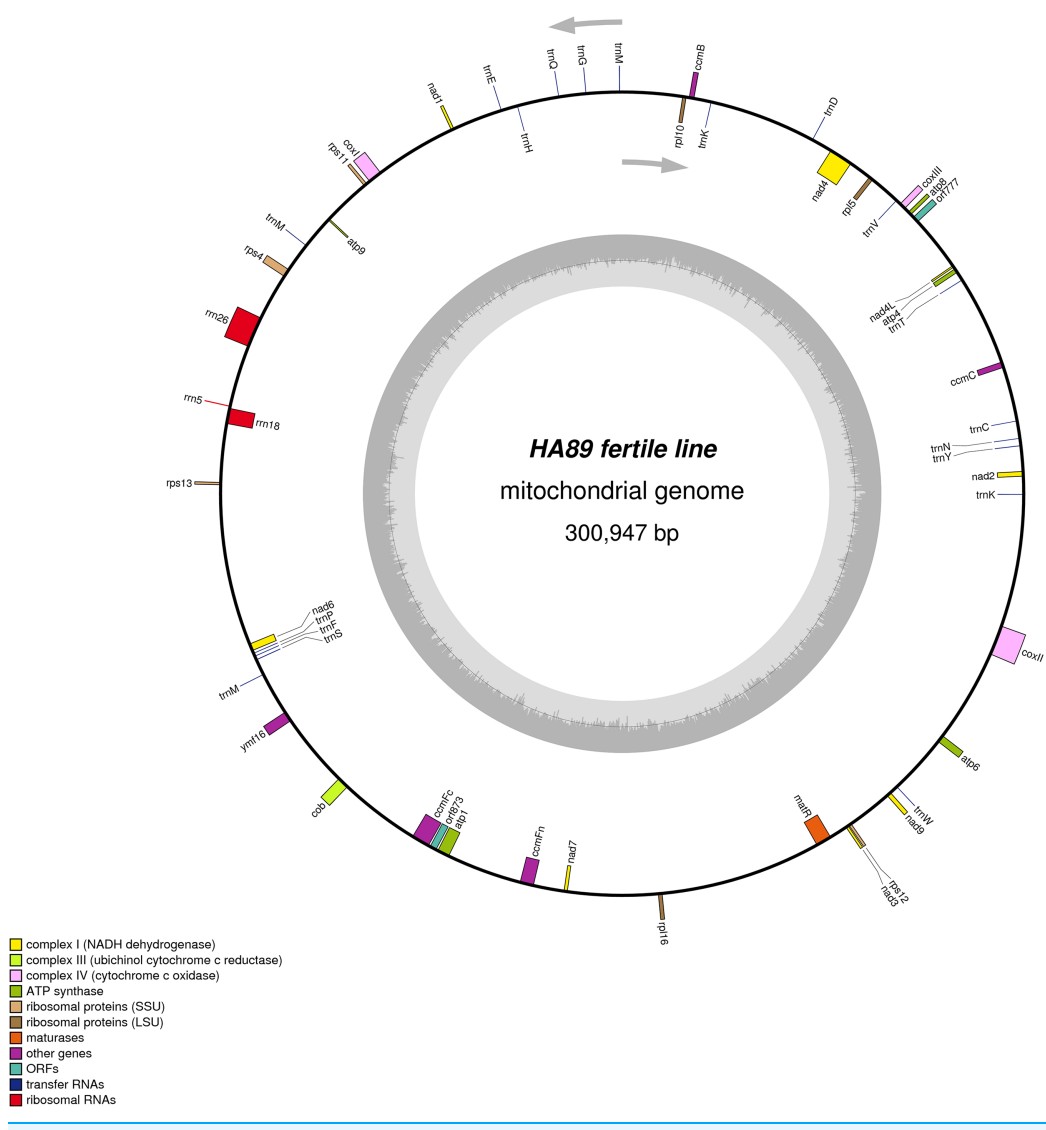

**Figure 1 Graphical mitochondrial genome maps of HA89 fertile line.**

## RESULTS AND DISCUSSION

### Organization of fertile line and PET1 mtDNA

De novo assembly of the fertile line and PET1 mitochondrial genomes revealed between nine and 12 large contigs (10–115 kb long), depending on $K$ value. The optimum $K$ value was 95. The obtained large contigs covered up to 95% of the reference genome. The remaining 5% of the mitochondrial genome correspond to repeats, therefore they were the breakpoints of contigs formation. Among numerous repeats in the mitochondrial genome, only one large (12,933 bp long) and six small (203–728 bp long) repeats played a crucial role in genome assembly and prevented single scaffold formation. Four regions (415–1,192 bp long) with 99% chloroplast DNA identity also affected the mitochondrial genome assembly. Eventually, manual assembly based on predominant

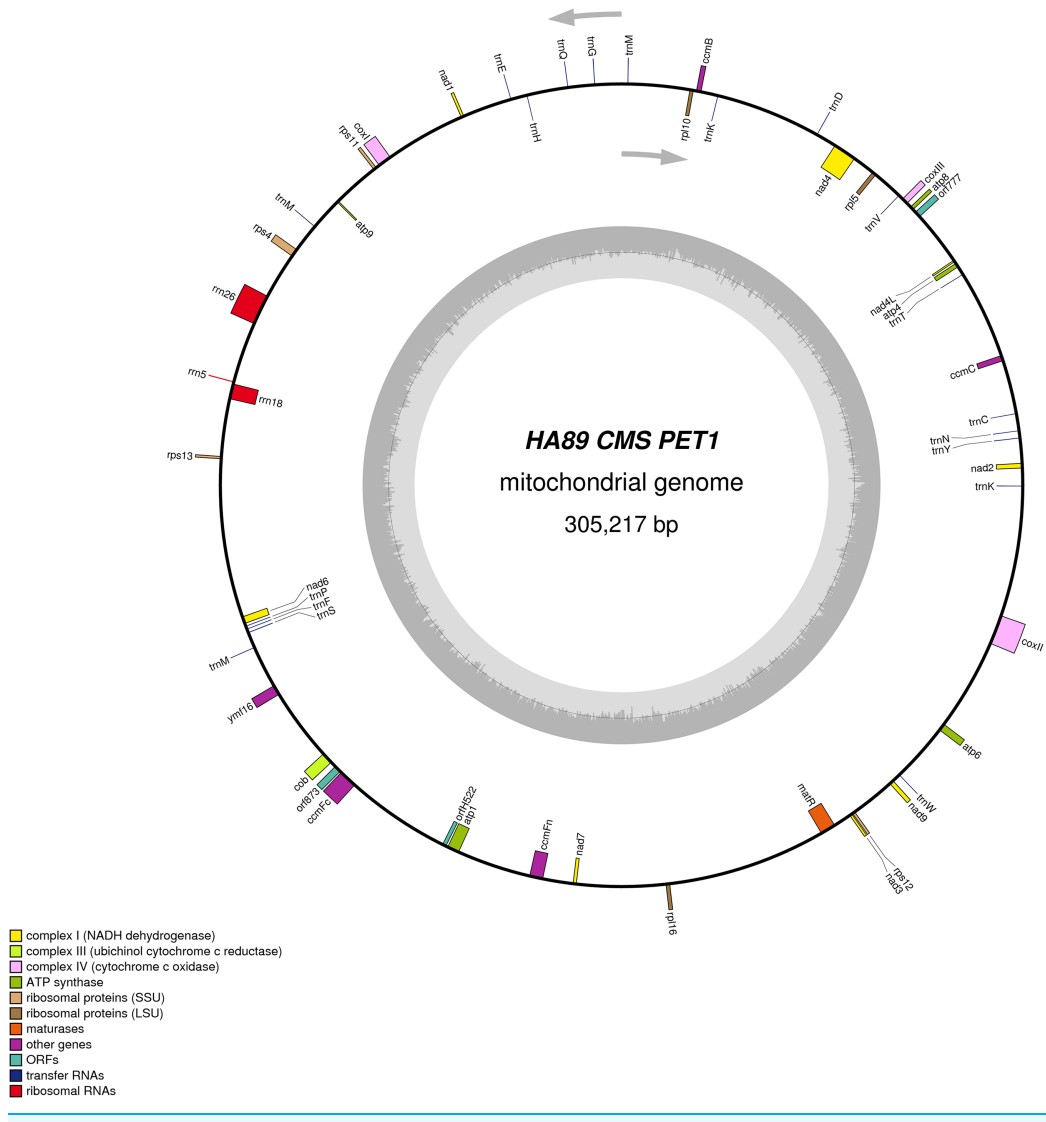

**Figure 2 Graphical mitochondrial genome map of HA89 (PET1) line.**

contigs obtained by SPAdes supplemented by analysis of reads alignment by Bowtie2 and validation of controversial regions performed by PCR analysis and Sanger sequencing, allowed summary sequencing of data in completed mitochondrial genomes. Circular mtDNA of HA89 and PET1 lines are presented in Figs. 1 and 2.

The mtDNA comparative analysis of sunflower fertile lines HA412 (NCBI accession NC_023337.1) and HA89 revealed two single nucleotide thymidine insertions: in positions 35690–35691 and 129368–129369 of NC_023337. Two SNP in the noncoding part of 301 kbp genome is a negligible difference. We did not amend the HA89 mitochondrion sequence in the NCBI GenBank and, in this paper, used the same positions of mtDNA for HA412 and HA89 lines for simplicity.

The PET1 mitochondrial genome had structural rearrangements as well as polymorphic sites compared to the fertile lines' (HA412/HA89) mtDNA. Previously,

using the restriction analysis and Sanger sequencing, large structural variations of mtDNA associated with the sterility of plants were detected in the PET1 CMS type of sunflower— a 11 kb inversion and a five kb insertion (*Kohler et al., 1991*). The results of the current study not only confirmed the presence of these reorganizations in the HA89 PET1 mitochondrial genome but also allowed detection of more precise genome changes: 11,852 bp inversion, 4,732 bp insertion, 451 bp deletion. The revealed insertion was 98% identical to the PET1 insertion that can be obtained from GenBank NCBI (accession Z23137.1). Comparing with the fertile line genome, we have identified 18 variants in HA89 (PET1) mtDNA. Among the nucleotide variations, eight were localized in SSR loci, two deletions (single and dinucleotide) and seven SNP, including one transition and six transversions, were predominantly located in noncoding regions (Table 2). The exceptions were nonsynonymous mutations in *orf777* (Asp251Glu), *nad6* (Ser232Tyr), *rpl16* (Lys32Gln). The HA89 (PET1) complete mitochondrial genome sequence has been deposited in the NCBI databank (accession MG735191).

## Organization of PET2 mtDNA

We assembled the PET2 mitochondrial genome following the described procedure that was used for identification of HA89 fertile and PET1 complete mtDNA sequences. The complete mitochondrion of HA89 (PET2) is 316,582 bp (Fig. 3) and, in comparison with the HA89 fertile line, contained large-scale reorganizations of the mtDNA structure as well as minor changes represented by variable sites. Among significant rearrangements, two translocations, two deletions and two insertions were determined.

Even in a single plant cell, the mitochondrial genome is represented by several DNA molecules with various structure (*Sloan et al., 2012*). The so-called "master chromosome"—the single mtDNA molecule is a rare type of mitochondrial genome organization (*Gualberto et al., 2014*). More often, a mitochondrion includes a set of sub-genomic forms (*Yang et al., 2015*). Because sub-genomes could form differing master chromosomes, the statement of translocations in plant mitochondrial genomes is equivocal. To compare complete mitochondrial genomes of HA89 (PET2) and fertile HA89 lines, two translocations of approximately 27.5 kb and 106.5 kb (positions 37,112–64,614 bp and 194,439–300,945 bp in HA89 fertile line mtDNA) could be established. Using specific PCR primers (Table 1) we could demonstrate that the mtDNA in sunflower can also form a 154-kb sub-genomic circle molecule (positions 36393–190650). In the sunflower genome there is a repeat region of 722 bp (36393–37114 = 189929–190650 positions of the fertile line) with 100% similarity, which makes the cyclization of sub-genome circular molecule possible. In the case of HA89 (PET2) this sub-genome circle molecule has the wrong insertion point in the genome as compared with the fertile analog and thus results in translocations appearance.

Deletion of 711 bp (35682–36393 positions in fertile line mtDNA) resulted in the absence of *orf777* in the HA89 (PET2) mitochondrial genome. The *orf777* codes for a putative protein with unknown function, which shows no similarities with other mitochondrial proteins. So its elimination can hardly be the molecular reason for CMS development. The other deletion of 3,780 nucleotides (190659–194439 positions)

**Table 2 Variation sites in mitochondrial DNA of HA89 CMS lines PET1 and PET2.**

| Position | Type | Fertile | PET1 | PET2 | Localization |
|---|---|---|---|---|---|
| 3031 | SSR | G5 | | **G6** | IGR *nad2-ccmC* |
| 3107 | SSR | T5 | | **T6** | IGR *nad2-ccmC* |
| 3275–3276 | INDEL | TA | | T**TT**A | IGR *nad2-ccmC* |
| 3281–3281 | INDEL | AT | | AT**T** | IGR *nad2-ccmC* |
| 4715 | SSR | T8 | | **T9** | IGR *nad2-ccmC* |
| 6207 | SSR | A8 | **A7** | **A7** | IGR *nad2-ccmC* |
| 6660 | SNP | A | | **G** | IGR *nad2-ccmC* |
| 7404 | SSR | G10 | | **G9** | IGR *nad2-ccmC* |
| 7919 | INDEL | A | | – | IGR *nad2-ccmC* |
| 9796 | SNP | T | | **C** | IGR *nad2-ccmC* |
| 10467 | SNP | A | | **C** | IGR *nad2-ccmC* |
| 10924 | SNP | A | | **C** | IGR *nad2-ccmC* |
| 12314 | SNP | T | | **C** | IGR *nad2-ccmC* |
| 19594 | SNP | G | | **A** | IGR *ccmC-atp4* |
| 23917 | SNP | G | | **T** | IGR *ccmC-atp4* |
| 31803 | SNP | A | | **C** | IGR *nad4L-orf259* |
| 34099 | SNP | A | | **C** | IGR *nad4L-orf259* |
| 34135–34136 | INDEL | AT | | AT**G**T | IGR *nad4L-orf259* |
| 34162 | SNP | T | | **C** | IGR *nad4L-orf259* |
| 35031 | SNP | C | | **A** | IGR *nad4L-orf259* |
| 35114 | SNP | C | | **A** | IGR *nad4L-orf259* |
| 35478 | SNP | T | | **C** | IGR *nad4L-orf259* |
| 35511 | SNP | G | | **A** | IGR *nad4L-orf259* |
| 35596 | SNP | G | | **C** | IGR *nad4L-orf259* |
| 36360 | SNP | T | **G** | – | *orf259* Asp251Glu |
| 42295 | SNP | C | | **A** | IGR *coxIII-rpl5* |
| 46039 | INDEL | A | – | | IGR *rpl5-nad4* |
| 49272 | SSR | C11 | **C9** | **C10** | IGR *nad4-ccmB* |
| 50856 | SNP | C | | **A** | IGR *nad4-ccmB* |
| 51678 | SSR | G10 | **G9** | **G9** | IGR *nad4-ccmB* |
| 62360 | SNP | T | | **G** | IGR *nad4-ccmB* |
| 62403 | SNP | G | | **A** | IGR *nad4-ccmB* |
| 63433–63434 | INDEL | TC | | T**C**C | IGR *nad4-ccmB* |
| 71497–71498 | INDEL | GT | | G**GGGC**T | IGR *rpl10-nad1* |
| 75332 | SNP | A | **C** | **C** | IGR *rpl10-nad1* |
| 91105 | SNP | G | | **T** | IGR *rpl10-nad1* |
| 91106 | SNP | A | | **C** | IGR *rpl10-nad1* |
| 105474 | SSR | T35 | | **T25** | IGR *nad1-coxI* |
| 108200 | SNP | T | | **G** | IGR *coxI-rps11* |
| 115915 | SNP | T | | **G** | IGR *atp9-rps4* |
| 116777 | SNP | G | **T** | | IGR *atp9-rps4* |

(Continued)

| Position | Type | Fertile | PET1 | PET2 | Localization |
|---|---|---|---|---|---|
| 119331 | SNP | G | | **A** | IGR *atp9-rps4* |
| 121108–121109 | INDEL | CC | | C**TT**C | IGR *atp9-rps4* |
| 122990 | SNP | A | | **C** | rps4 (synonymous) |
| 133546 | SNP | T | | **A** | IGR *rrn26-rrn5* |
| 133547–133548 | INDEL | AT | | A**GG** | IGR *rrn26-rrn5* |
| 133548 | SNP | T | | **G** | IGR *rrn26-rrn5* |
| 133549 | SNP | A | | **C** | IGR *rrn26-rrn5* |
| 156213 | SNP | C | | **A** | IGR *rps13-nad6* |
| 156621–156622 | INDEL | CC | | C**CTA**C | IGR *rps13-nad6* |
| 157459 | SNP | T | | **G** | IGR *rps13-nad6* |
| 169028 | SNP | G | **T** | | *nad6 (Ser232Tyr)* |
| 170185 | SSR | T14 | **T12** | **T12** | IGR *nad6-ymf16* |
| 174932–174933 | INDEL | AC | | A**CTCGACTGAAA GGAAAGGTAC GAAGTGG**C | IGR *nad6-ymf16* |
| 175179 | SNP | G | | **T** | IGR *nad6-ymf16* |
| 178406 | SSR | T9 | **T8** | | *ymf16* intron |
| 184739 | SSR | A10 | **A11** | | IGR *ymf16-cob* |
| 188363 | SSR | T11 | **T10** | **T10** | *cob* intron |
| 189980 | SNP | G | | **T** | IGR *cob-ccmFc* |
| 195008 | SNP | G | | **T** | IGR *cob-ccmFc* |
| 195015 | SNP | C | | **A** | IGR *cob-ccmFc* |
| 200174 | SNP | G | | **A** | *ccmfC* intron |
| 200515 | SNP | G | | **A** | *ccmfC* intron |
| 202672 | SNP | T | **C** | | IGR *orf873-atp1* |
| 204990 | SNP | C | | **A** | IGR *atp1-ccmFn* |
| 204846–204847 | INDEL | AA | | A**T**A | IGR *atp1-ccmFn* |
| 207965 | SSR | G10 | | **G12** | IGR *atp1-ccmFn* |
| 209335–209336 | INDEL | AA | – | | IGR *atp1-ccmFn* |
| 209458 | SNP | G | | **A** | IGR *atp1-ccmFn* |
| 212638 | SSR | C9 | | **C12** | IGR *atp1-ccmFn* |
| 215916 | SNP | C | | **T** | IGR *ccmFn-nad7* |
| 223917 | SNP | A | | **C** | IGR *nad7-rps3* |
| 223925–223926 | INDEL | GA | | GA**A** | IGR *nad7-rps3* |
| 226977–226978 | INDEL | AC | | A**CGTTGTTTT**C | IGR *nad7-rps3* |
| 230112 | SNP | A | **C** | | *rpl16* (Lys32Gln) |
| 232826 | SNP | G | | **T** | IGR *rpl16-matR* |
| 239880 | SNP | G | | **A** | IGR *rpl16-matR* |
| 239988 | SNP | A | | **C** | IGR *rpl16-matR* |
| 241035 | SNP | G | | **A** | IGR *rpl16-matR* |
| 241475 | SNP | A | | **C** | IGR *rpl16-matR* |
| 246053 | SNP | C | | **T** | IGR *rpl16-matR* |

| Table 2 (continued). | | | | | |
|---|---|---|---|---|---|
| Position | Type | Fertile | PET1 | PET2 | Localization |
| 248266 | SSR | A14 | **A10** | **A9** | IGR *rpl16-matR* |
| 249347 | SSR | T8 | | **T9** | IGR *rpl16-matR* |
| 249361 | SNP | C | | **A** | IGR *rpl16-matR* |
| 260901 | SNP | G | | **T** | IGR *nad9-atp6* |
| 262080 | SNP | G | | **A** | IGR *nad9-atp6* |
| 269062 | SNP | G | **C** | **C** | IGR *nad9-atp6* |
| 269134 | SNP | A | | **C** | *atp6* (synonymous) |
| 270676 | SNP | G | | **T** | IGR *atp6-coxII* |
| 273344 | SNP | C | | **A** | IGR *atp6-coxII* |
| 276834 | SNP | T | | **G** | IGR *atp6-coxII* |

**Note:**

Nucleotide positions are specified according to fertile line mtDNA. IGR—an intergenic region. In case of indels the deletions are indicated as "–" and the inserted nucleotide are in bold.

affects only noncoding sequences. Notably, the deletions are associated with sub-genome integration regions (36393–37114 and 189929–190650 positions). So there is a possibility, that deletions in these regions could impair master chromosome assembly, which results in the translocations formations, as described above.

More significant mtDNA reorganizations are two revealed insertions—5,050 bp and 15,885 bp. Among them, the 5,050 bp insertion is most likely not involved in the origin of the CMS phenotype. This insertion was found in the intergenic region *atp6-cox2* (275230–275231 positions of fertile line mtDNA), and it does not lead to the formation of new ORFs directly in the place of insertion into the mitochondrial genome. Moreover, there are no large (more than 300 nucleotides) ORFs within the sequence of 5,050 bp insertion. The exception is *orf645* putatively translated into a polypeptide of 215 amino acids. Twenty-three amino acids at the N-terminus of this 215 amino acid protein were similar to N-terminus of ribosomal protein S3. We determined transcripts of *orf645* in PET2 CMS line by qPCR, using specific primers (Table 1). However, mRNA of *orf645* was absent in the fertile line and PET1 CMS line. Most often the molecular reason for CMS phenotype development is the emergence of chimeric ORFs with transmembrane domains, such as ATPase complex subunits, respiratory-related proteins, etc. (*Gillman, Bentolila & Hanson, 2007*; *Yang, Huai & Zhang, 2009*). Consequently, even if *orf645* is translated *in vivo*, its role in male sterility development most likely is negligible, as there are no similarities with respiratory/ATP synthesis-related proteins.

The 15,885 bp insertion in the intergenic region *nad4L-ccmFc* is more complicated than the 5,050 bp insertion and includes different ORFs. First of all, it should be highlighted that most of the insertion (9,482 bp) has 100% similarity to another PET2 mtDNA region (126260–135741 positions). The repeating part of the insertion could be divided into two parts—6,097 bp (35686–41782 positions of PET2 mitochondrion) are common (99–100% identity) for *Helianthus* mitochondrion: 269147–275243, 273418–279514 and 126260–132356 positions of fertile line, PET1 and PET2 CMS lines, respectively. Such a repeat predominantly consists of noncoding sequences, except *atp6* gene. The other

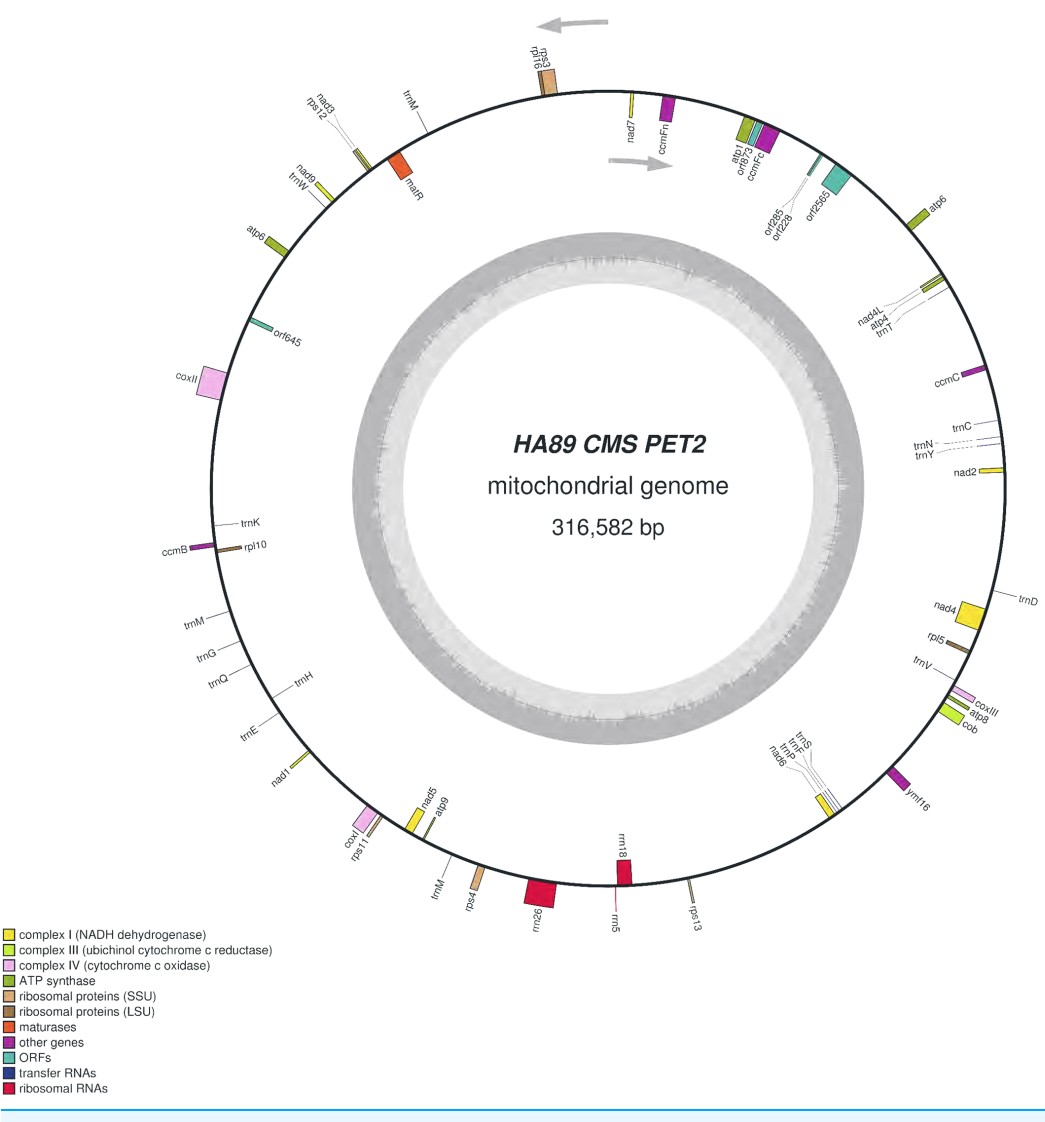

**Figure 3 Graphical mitochondrial genome map of HA89 (PET2) line.**

3,385 bp (41783–45167 positions of PET2 mitochondrion) have 100% similarity to the part of 5,050 bp insertion (132357–135741 positions of PET2 mitochondrion). However, this part of the insertion does not contain coding sequence. The next part of the 15,885 bp insertion counts 4,849 nucleotides which are unique for the PET2 mitochondrion (45168–50016 positions). The *orf2565* is encoded in this part of insertion. The last 1,554 bp of the 15,886 bp insertion (50017–51570 positions of PET2 mitochondrion) have complex origin. A total of 390 nucleotides of this insertion (50017–50406 positions of PET2 mitochondrion) are similar to 114179–114568 positions of fertile line mtDNA, the next 271 bp (50406–50676 positions) are unique, then 500 bp (50677–51177 positions) are complement to 114587–115087 positions of the HA89 fertile mitochondrion and the rest 393 nucleotides (50677–51570) are also unique. From a functional point this region presents duplication of *atp9* gene combined with 271 bp

insertion and deletion of 12 bp, which resulted in two new ORFs formation—*orf228* and *orf285* (*Reddemann & Horn, 2018*). So the 15,885 bp insertion, especially its last 1,554 nucleotides, is the most important mitochondrial genome rearrangement, probably associated with PET2 CMS phenotype. It is also notable that 15,885 bp insertion has the same region in mtDNA of PET2 as the other reorganizations—711 and 3,780 bp deletions and 106.5 kb translocation.

Summarizing the functional changes data obtained, the 15,885 bp insertion consists of four coding sequences: duplicated *atp6* gene and three new ORFs—*orf2565, orf228* and *orf285*. *Atp6* chimeric ORFs or new ORFs co-transcribed with *atp6* are the quite common causes of CMS development in different plant species (*Kim, Kang & Kim, 2007*; *Jing et al., 2012*; *Tan et al., 2018*). So we proposed that *atp6* gene and its colocalized area are of particular interest as the candidate sequence for CMS phenotype development. The annotation of *H. annuus* L. mitochondrion (NC_023337.1) presents the *atp6* gene coding mRNA which consists of 906 nucleotides. But the fact is that in the 5′ adjoining sequence (150 bp) to *atp6* start codon there is the other one initiating codon (ATG). Transcription from this codon results in new mRNA counting 1,056 nucleotides. The elongated transcript could be translated in putative ATP6 protein with additional 50 amino acids, wherein 37 of 50 amino acids are identical with N-terminus of coxI. The similar extension of protein was discovered in another CMS type of sunflower—ANT1 (*Spassova et al., 1994*), but the described protein had additional 87 aa at the C-terminus of the ATP6 protein. So we assumed that extended transcript, which could be produced in *atp6* duplicated region, may cause CMS phenotype. To verify the hypothesis, the expression levels of *atp6* and 5′ elongated *atp6* transcripts were analyzed using qPCR with the same reverse primer but different forward primers (Table 1). The elongated *atp6* transcript has been expressed at the same level as the *atp6* gene (Gene ID: 18250997) both in PET1, PET2 CMS and fertile lines. So the *atp6* transcript counting 1,056 nucleotides is the normal one, and there is a mistake in its annotation in NC_023337.1. Moreover, the *atp9* gene also (Gene ID: 18250970) has wrong transcript annotation—it counts only 261 bp instead of proper one with 300 bp. Using qPCR and specific primers (Table 1) we established the same expression level for 261 bp and 300 bp transcripts in all three studied lines. Notable that the relative expression level ($\Delta Ct$) of *atp6* gene to *atp1* gene in PET2 CMS line had no significant difference as compared with fertile and PET1 CMS analogs, despite *atp6* duplication in PET2 mitochondrion. Thereby *atp6* duplication in PET2 mitochondrion does not involve in CMS appearance.

All three revealed ORFS (*orf2565, orf228* and *orf285*) had demonstrated transcription activity in PET2 CMS line but not in the fertile or PET1 CMS analogs. The *orf2565* translates to the putative polypeptide of 855 amino acids. Homology search in the NCBI database using BLAST pointed to similarity to DNA polymerase (type B). The impact of *orf2565* on CMS phenotype development is quite doubtful, taking into account that the polypeptide has no transmembrane domains. The *orf228* encodes a polypeptide of 76 aa, of which 75 are identical to C-terminus of ATPase subunit 9, as well as a new start codon (AUG) is formed due to the 271 bp insertion. Naturally, ATPase subunit 9 has two transmembrane domains—near N- and C-terminus, in *orf228* there are also two

predicted transmembrane domains (*Reddemann & Horn, 2018*). However, it should be noted that the N-terminus transmembrane domain of *orf228* encoded polypeptide lacks four amino acids as compared with *atp9* ones. This difference, in turn, could affect polypeptide (*orf228*) interaction with mitochondrial membrane and so results in mitochondrial membrane potential changes. The *orf285* encodes a polypeptide of 95 aa, of which 18 are complement to N-terminus of ATPase subunit 9. In total *orf228* and *orf285* polypeptides share 93 of 99 amino acids of the *atp9* protein. The polypeptide encoded by *orf285* also has the transmembrane domain which formed by predominantly 20–42 aa. Thus both ORFs (*orf228, orf285*) could be the main reason the development of PET2 CMS type.

The comparative analysis of PET2 mitochondrion sequence with complete mtDNA of fertile line revealed 83 polymorphic sites—14 SSR, 13 small indels (1–29 bp) and 56 SNP (Table 2). Among nucleotide variations only two (synonymous) SNP were in the coding sequence of genes *rps3* and *atp6*. Additional analysis of distribution of variants can offer insight into functional properties and evolution of sunflower mtDNA (*Triska et al., 2017*). Interestingly, PET1 and PET2 share 10 polymorphic sites as compared with the fertile line. The obtained HA89 (PET2) mitochondrial genome sequence has been deposited to the NCBI GenBank (accession MG770607.2). It is important to note that the sets of primers used for identification of PET2 insertions may be used for designing molecular markers for this type of CMS in sunflower.

## CONCLUSIONS

Comparative analysis of HA89 fertile and PET1 CMS analog mitochondrial genomes revealed 11,852 bp inversion, 4,732 bp insertion, 451 bp deletion and 18 variant sites. In the mtDNA of HA89 (PET2) CMS line we determined 27.5 kb and 106.5 kb translocations, 711 bp and 3,780 bp deletions, as well as, 5,050 bp, 15,885 bp insertions and 83 polymorphic sites. From a functional point of view, there is the elimination of *orf777*, duplication of *atp6* gene and appearance of four new ORFs with transcription activity specific for the HA89 (PET2) CMS line—*orf645*, *orf2565*, *orf228* and *orf285*. We hypothesize that *orf228* and *orf285* could be the main reason for the development of PET2 CMS phenotype, while the contribution of other mtDNA reorganizations in CMS formation is negligible.

## ACKNOWLEDGEMENTS

We thank reviewers and the editor for their insightful suggestions and comments on the paper, which has helped us to improve the manuscript significantly. We are grateful to the cooperators of Northern Crop Science Laboratory (Fargo, USA) for providing seeds of HA89 alloplasmic lines of sunflower to the genetic collection of the N. I. Vavilov Institute of Plant Genetic Resources.

### Funding

This research was supported by the Ministry of Education and Science of Russia project no. 6.929.2017/4.6 and Russian Science Foundation project no. 14-50-0002. Analytical

work was carried out on the equipment of centres for collective use of Southern Federal University "High Technology." The funders had no role in study design, data collection and analysis, decision to publish, or preparation of the manuscript.

## Grant Disclosures

The following grant information was disclosed by the authors:
Ministry of Education and Science of Russia: 6.929.2017/4.6.
Russian Science Foundation: 14-50-0002.

## Competing Interests

The authors declare that they have no competing interests.

## Author Contributions

- Maksim S. Makarenko conceived and designed the experiments, performed the experiments, analyzed the data, contributed reagents/materials/analysis tools, prepared figures and/or tables, authored or reviewed drafts of the paper, approved the final draft.
- Igor V. Kornienko performed the experiments, contributed reagents/materials/analysis tools.
- Kirill V. Azarin performed the experiments, analyzed the data, prepared figures and/or tables.
- Alexander V. Usatov conceived and designed the experiments, analyzed the data, contributed reagents/materials/analysis tools.
- Maria D. Logacheva performed the experiments, analyzed the data, contributed reagents/materials/analysis tools, authored or reviewed drafts of the paper.
- Nicolay V. Markin analyzed the data, prepared figures and/or tables.
- Vera A. Gavrilova conceived and designed the experiments, contributed reagents/materials/analysis tools.

## DNA Deposition

The following information was supplied regarding the deposition of DNA sequences:
The complete mitochondrion sequence of CMS line HA89(PET1) has been deposited in GenBank (the accession number MG735191). The complete mitochondrion sequence of CMS line HA89(PET2) has been deposited in GenBank (accession MG770607.2).

## Data Availability

Makarenko, Maksim; Logacheva, Maria (2018): NGS reads of sunflower line HA89. figshare. Fileset. DOI 10.6084/m9.figshare.5787171.v1.

## Supplemental Information

Supplemental information for this article can be found online at http://dx.doi.org/10.7717/peerj.5266#supplemental-information.

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
