# Peer review of "Mitochondrial genomes organization in alloplasmic lines of sunflower (Helianthus annuus L.) with various types of cytoplasmic male sterility"

_PeerJ, doi:10.7717/peerj.5266_

## Round 0.1 · original submission · Major Revisions

Both reviewers asked to have PET2 redone. I believe, this must be done for the publication.

Reviewer 1 ·

Basic reporting

The manuscript is well written and describes the assembly of the mitochondrial DNA of the fertile, PET1 and PET2 cytoplasm in sunflower from next generation sequencing data, which in principle is very interesting and new for PET1 and PET2. However, the assembly of PET2 is not satisfactory. The offer of two circles that cannot be combined is not helpful. It indicates the mtDNA for PET2 is not correctly assembled.

Experimental design

Results for the RT-PCR should be shown and should include the study of fertility restored hybrids.

Validity of the findings

1. The manuscript requires presentation of one circle for PET2. The coverage might not be high enough or hybridization results might be helpful to solve the problem. The early publication of two premature mtDNA circles for PET2 is not helpful.
2. For claiming an involvement with the CMS phenotype at least the effect on transcripts in fertility-restored hybrids should be shown. It is not enough that an orf is expressed in the male sterile line, but not the fertile line.
3. The sequence of the 5 kb insertion in PET1 is already present in NCBI (accession Z23137.1) and should be mentioned in the manuscript. Homology between the sequences should be discussed.
4. The two open reading frames, orf645 and orf1053, which are supposed to be involved in the CMS phenotype encode proteins of about 21 kDa and about 35 kDa, respectively. The CMS-specific protein detected by in organello translation for PET2 has a size of 12.4 kDa (Horn and Friedt 1999). This might be an indication that both new open reading frames, only transcription is stated for orf645, might not be involved.

Additional comments

Please take your time to correctly assemble the PET2 mtDNA! I think you do not gain anything by publishing preliminary data even though you put already a lot of work into the the manuscript.

·

Basic reporting

The paper should be seen by a native English speaker or a technical editor. There are multiple typos and unnecessary punctuation signs. For example, " revealed two single nucleotide insertion: thymidine insertions in 35690-35691 and 129368-129369 positions of NC_023337" should be " revealed two single nucleotide thymidine insertions: in positions 35690-35691 and 129368-129369 of NC_023337". And the following sentence " Considering that due to 2 SNP is not reasonable for appending current data (HA89 fertile line complete mitochondrion sequence) in the NCBI databank" should also be completely reworked. "MC2 cyclization was failed" -> :"MC2 cyclization has failed"

Experimental design

The PET2 genome used is not fully assembled. It would be desirable to produce the entire genome. I do not anticipate this task to be theoretically or practically challenging

Validity of the findings

There was a new open reading frame orf1053 described in the genome. This in silico discovery should be supported by RNA seq data. This may be a chimeric gene or assembly artifact,etc. The authors need to analyze structure of promoters and distribution of TBFS to support the function of the identified genes.

---

## Round 0.2 · Minor Revisions

Dear authors,

We are close to accepting the paper but there are a number of notes of the Reviewer 2, both of scientific and editorial nature. Please address them carefully in the next version of the manuscript.

with best regards

Reviewer 1 ·

Basic reporting

I think the manuscript has been significantly improved. The circular assembly of PET2 mtDNA looks much better now. It represents a unique contribution and should therefore be published. The other two assemblies for the fertile and the CMS PET1 are also of public interest. In the latter two it would be nice if the nad5 would be given with its location. The PET2 DNA assembly also supports the data obtained by Reddemann, Horn 2018. Orf228 represents orf231 without counting the stop codon and orf246 corresponds to orf288 (without stop codon and the possible elongation at the 5’ end of the atp9 gene). However, in our fragments nad5 was on the same fragment as atp9 and the split atp9 was on the fragment together with the orf2565 (DNA polymerase type b homology). This might be looked into it again.

Experimental design

no comment

Validity of the findings

Next generation sequencing data allow assembly of whole mtDNA genomes, which has been done in this manuscript. The coverage was sufficient for obtaining the results. Duplications were detected.

Additional comments

In Table 1 amplicon size needs to be corrected. Either use bp than some values have to be changed from 0.9 to 900 or use kbp but then other values have to be changed from 162 to 0.162. Please correct one way or the other!
The manuscript still needs some improvement with regard to the English language. Some corrections are suggested here and should be made in the manuscript:
Line 17: is often associated
Line 18-19: Investigations of the CMS phenomenon also promote understanding of fundamental
issues of
Line 21: Helianthus annuus in italics
Line 45-46: large sizes….. sequences which can result in complex sub-genomic structures
Line 52: Leclercq in an interspecific
Line 55: 11-kb-inversion and a 5-kb-inversion
Line 56: Those rearrangements
Line 61+62: 16-kDa-protein
Line 67: Interestingly, stable transgenic
Line 72: restorer
Line 78: Leclercq (1969)
Line 113: from five plants
Line 186: which has already been available in
Line 192: in the NCBI databank
Line 197: both, large-scale
Line 199: one translocation, two deletions and three insertions were determined.
Line 208: we could demonstrate that the mtDNA in sunflower can also form a 154-kb sub-genomic
circular molecule
Line 211: makes the cyclization of ……..molecule possible. …………….sub-genomic circular molecule
Line 215+216: The orf777 codes for a putative …..function, which shows no similarities
Line 218: ……..noncoding sequences. Notably the insertions
Line 221: …..resulted in a translocation, as described above.
Line 223: … most likely not involved in the origin of the CMS phenotype.
Line 228: orf645 putatively translated into a polypeptide……. Twenty-three amino acids at the N-
terminus
Line 230: transcripts of orf645 in PET2 CMS line by qPCR, using specific primers (Table
1). However, mRNA of orf645
Line 241: for Helianthus (in italics) mitochondrion
Line 246: this part of the insertion …… coding sequences.
Line 252: ….domains. The opposite is the case for the atp6 gene.
Line 264+265: at the C-terminus of the ATP6 protein.
Line 269: nearly
Line 270: atp6 in italics
Line 271: Interesting to note is that
Line 276: probably associated
Line 277+278: is a 1558-bp-insertion localized between the genes nad5 and ccmFc. The 14330-bp-
insertion as well as the 1558-bp-insertion have a complement to another part of the
genome and a unique sequence.
Line 283: in two new ORFs
Line 284+285: Orf228 encodes a polypeptide of 76 aa, of which 75 are identical
Line 286: resulting
Line 286: …..lacks four amino acids
Line 293: Add: The orf246 corresponds to orf288 (Reddemann, Horn 2018), which takes a possible
elongation at the 5’-end of the atp9 gene in sunflower into account.
Line 299: Interestingly PET1

·

Basic reporting

The authors fixed the majority of the typos and significantly improved presentation of the material. Mathematical formulas (such as 1+E^ΔCt formula) should be typed using an appropriate equation editor.

Experimental design

The authors assembled the genome and, therefore, the manuscript does not look like a dump of half-cooked results anymore

Validity of the findings

qPCR and Sanger validation of discovered rearrangements was conducted.

Additional comments

The paper was significantly improved. The authors took all suggestions into consideration. I recommend to accept the manuscript in its current form, except for minor formatting changes as described above.

---

## Round 0.3 · accepted · Accept

Dear Maksim,

We are accepting the paper. Please, take care to carefully proofread the manuscript.

#